# A Systematic Review of Cerebral Functional Near-Infrared Spectroscopy in Chronic Neurological Diseases—Actual Applications and Future Perspectives

**DOI:** 10.3390/diagnostics10080581

**Published:** 2020-08-12

**Authors:** Augusto Bonilauri, Francesca Sangiuliano Intra, Luigi Pugnetti, Giuseppe Baselli, Francesca Baglio

**Affiliations:** 1Department of Electronics, Information and Bioengineering, Politecnico di Milano, 20133 Milan, Italy; augusto.bonilauri@polimi.it (A.B.); giuseppe.baselli@polimi.it (G.B.); 2IRCCS Fondazione Don Carlo Gnocchi ONLUS, CADITER, 20148 Milan, Italy; lpugnetti@dongnocchi.it (L.P.); fbaglio@dongnocchi.it (F.B.); 3Faculty of Education, Free University of Bozen-Bolzano, 39100 Bolzano, Italy

**Keywords:** neurovascular coupling, fNIRS, neurological disease, neuroimaging, Parkinson’s Disease, Alzheimer’s Disease, Mild Cognitive Impairment

## Abstract

Background: The management of people affected by age-related neurological disorders requires the adoption of targeted and cost-effective interventions to cope with chronicity. Therapy adaptation and rehabilitation represent major targets requiring long-term follow-up of neurodegeneration or, conversely, the promotion of neuroplasticity mechanisms. However, affordable and reliable neurophysiological correlates of cerebral activity to be used throughout treatment stages are often lacking. The aim of this systematic review is to highlight actual applications of functional Near-Infrared Spectroscopy (fNIRS) as a versatile optical neuroimaging technology for investigating cortical hemodynamic activity in the most common chronic neurological conditions. Methods: We reviewed studies investigating fNIRS applications in Parkinson’s Disease (PD), Alzheimer’s Disease (AD) and Mild Cognitive Impairment (MCI) as those focusing on motor and cognitive impairment in ageing and Multiple Sclerosis (MS) as the most common chronic neurological disease in young adults. The literature search was conducted on NCBI PubMed and Web of Science databases by PRISMA guidelines. Results: We identified a total of 63 peer-reviewed articles. The AD spectrum is the most investigated pathology with 40 articles ranging from the traditional monitoring of tissue oxygenation to the analysis of functional resting-state conditions or cognitive functions by means of memory and verbal fluency tasks. Conversely, applications in PD (12 articles) and MS (11 articles) are mainly focused on the characterization of motor functions and their association with dual-task conditions. The most investigated cortical area is the prefrontal cortex, since reported to play an important role in age-related compensatory mechanism and neurofunctional changes associated to these chronic neurological conditions. Interestingly, only 9 articles applied a longitudinal approach. Conclusion: The results indicate that fNIRS is mainly employed for the cross-sectional characterization of the clinical phenotypes of these pathologies, whereas data on its utility for longitudinal monitoring as surrogate biomarkers of disease progression and rehabilitation effects are promising but still lacking.

## 1. Introduction

The ageing of the worldwide population occurring in the last decade is mainly due to the improvements in the quality of life and the rise of life expectancy [1], which has globally risen from an average of 65.6 years in 1990 to 73 years in 2017 [2]. At the same time, a decrease of healthy life expectancy and an increase of disability due to non-communicable diseases—so called adjusted life-expectancy—has occurred [2]. The Global Burden of Diseases, Injuries and Risk Factors Study highlights a high variability in the pattern of age-related disease burden among countries and, according to the World Health Organization, it suggests that the percentage of unhealthy older people will increase with time [3]. Age-related diseases represent about the 51.3% of the total global disease burden [4] and, among the most common chronic neurological conditions include Parkinson’s disease (PD), Multiple Sclerosis (MS), Mild Cognitive Impairment (MCI) and Alzheimer’s Disease (AD). Moreover, the economic impact of age-related pathologies on healthcare systems is emerging as a concrete issue [5]. In particular, the management of patients affected by chronic non-communicable neurological diseases requires the adoption of longitudinal intervention strategies. In this perspective, the importance of rehabilitation is growing because of its ability to increase the quality of life of individuals in the aged population [6,7]. However, the implementation of targeted and cost-effective strategies, tailored to both clinical and individual needs, is still unsatisfactory [8], not to mention the need to monitor the effectiveness of rehabilitation both in terms of individual changes of physical condition, social and psychological welfare [9,10] In this framework, the present work focuses on methods able to follow the outcomes of interventions to restore physical impairments, in which neuroplasticity (i.e., the ability of nervous system to reorganize its structural and functional aspects following a repetitive intrinsic or extrinsic stimulation) represents one of the main mechanisms called into play [11,12]. In principle, this mechanism allows us to indirectly relate the progression of the disease with actual modifications occurring in the brain [13] and objective instrumental measurements can more effectively drive the planning of individualized rehabilitation interventions.

The monitoring of rehabilitation programs is generally accomplished by comparing variations of standard clinical scales and neuropsychological outcome measures. In this perspective, functional neuroimaging techniques—including magneto/electroencephalography (MEG/EEG), radiotracer imaging and functional magnetic resonance (fMRI)—have provided objective methods to investigate cerebral structure, as well as to quantify functional alterations in neurological diseases [14]. Besides, the objective quantification of physiological and biomolecular alterations assume an important role for monitoring the progression of either structural, functional or molecular transformations related to clinical outcome measures [15]. More recently, the integration of different neuroimaging techniques and clinical outcomes have provided additional insights into the phenotypical characteristics of neurological disorders. For example, it has been reported that different neurological conditions can be differentiated according to gait variability [16,17], suggesting that this measure may be also employed to evaluate the efficacy of specific rehabilitative interventions. Unfortunately, the identification and the choice of specific rehabilitation outcome measures remain controversial in research and clinical practice [18]. Therefore, systematic neurophysiological correlates of cerebral activity and neuroplasticity mechanisms during rehabilitation and clinical treatments are not always available.

According to the principle of neurovascular coupling, fMRI employs blood oxygenation level dependent signal as an indirect biomarker of brain activity and is widely accepted as the gold standard modality of functional neuroimaging techniques [19]. However, the application of neuroimaging methods, including fMRI, is often limited either by high costs, long scan times, poor temporal resolution and sensitivity to motion that reduce patient compliance and the possibility to assess responses evoked by complex stimuli. A more versatile solution to investigate cerebral activity that is gaining an increasing relevance is functional Near-infrared Spectroscopy (fNIRS) [20]. It is a non-invasive and non-ionizing optical imaging technique that employs near-infrared (NIR) light to measure cortical oxygenation and consequently cortical activations. This technique employs pairs of NIR sources and detectors placed over predefined scalp locations to estimate light intensity attenuations that are then converted to chromophore concentrations according to the modified Beer-Lambert law [21]. The chromophores of interest are represented by oxygenated (HbO2) and deoxygenated hemoglobin (HbR), since the light absorption of other major components of biological tissues are essentially stationary and allow sufficient transmittance in the NIR region. Accordingly, Continuous Wave fNIRS can only quantify relative variations in fNIRS [22], while more sophisticated dynamic analyses are required for absolute concentrations of HbO2 and HbR by means of either Frequency Domain or Time Domain fNIRS [23]. Nowadays, Continuous Wave fNIRS is the main implemented and commercially available solution in actual clinical instrumentations, while the application of Frequency or Time Domain technologies is still limited by their higher costs and complexity but do represent a promising approach for developing more robust and accurate optical brain imaging methods [23,24,25]. Henceforth, the term fNIRS will refer to applications of this technique for measuring task-evoked hemodynamic response, while the term NIRS will more specifically indicate cortical oxygenation quantification.

fNIRS is better suited to performing multiple acquisitions in rehabilitation settings because it allows us to overcome some limitations of other more traditional neuroimaging technologies. Indeed, this technique provides low-cost functional measurements in an open environment without causing excessive discomfort to the subject [26] nor does it interfere with diagnostic, therapeutic and pharmacological procedures. In addition, fNIRS can be operated during the execution of a wide variety of different tests, such as motor, somatosensory or cognitive tasks [27]. As well, the assessment of postural control and free-walking can also be carried out, thus allowing to investigate different neurological disorders in more ecological conditions [17,28]. Moreover, fNIRS is less sensitive to motion artifacts and can be employed with study patients that are less compliant to other neuroimaging modalities such as fMRI [29]. This technique is also based on the principle of neurovascular coupling exploited in fMRI [25] though with a lower spatial resolution, since penetration depth is mainly determined by the source-detector separation distances over subject’s scalp [30]. Indeed, before reaching the cortical surface the path of light from source to detector is affected by extra-cerebral and subcutaneous tissues which have heterogeneous optical properties that affect the overall changes in the absorption spectrum [31]. As a result, measurements can be affected by non-evoked neuronal activity and systemic physiological interferences such as heart rate (~1 Hz), breathing rate, (~0.3 Hz), Mayer waves (~0.1 Hz), very low frequency oscillations related to vessels neurogenic activity (~0.01 Hz) and endothelial function (~0.07 Hz), which may give rise to false positives and negatives in cortical activation and demand careful pre-processing of fNIRS data [20,32,33]. Importantly, fNIRS can target only the cortical areas adjacent to the skull [34] and is therefore blinded to medial cortical areas and to subcortical structures that are conversely mapped by volumetric fMRI.

Nowadays, fNIRS is primarily used for research purposes. In the field of clinical neurology and neurorehabilitation, this technology is mostly employed as a monitoring tool of functional recovery in stroke [35] and for therapeutic applications associated with brain-computer interface and neuro-feedback [26,36]. For example, because motor learning is known to induce cortical reorganization that is correlated with functional gains [37], cortical activation has been proposed as a potential biomarker in balance recovery interventions [38]. Moreover, it also correlates with mirror therapy success [39] and can be associated with noticeable and immediate rehabilitation-related effects [40]. Together with standard clinical outcome measures and other performance measures for the assessment of rehabilitation in patients affected by neurological disorders, fNIRS will potentially provide an additional valuable means to monitor neuroplasticity.

Under the above perspectives, we have undertaken a systematic review of actual cerebral fNIRS applications in the most common chronic neurological conditions of old age, namely PD, MCI and AD, focusing on motor and cognitive disability. We also reviewed applications in MS as the most common neurological disorder in young adults. With respect to stroke recovery, there is a paucity of longitudinal studies assessing the effects of rehabilitation programs and intervention strategies where fNIRS is systematically employed for assessing cerebral hemodynamic responses. Nonetheless, the studies reviewed here provide insights and suggestions for future applications of this technique to monitor the effects of rehabilitative interventions.

## 2. Methods

We provide a systematic review of published peer-reviewed original research addressing fNIRS for the investigation of functional cerebral activity in PD, MS, AD and MCI patients. The literature search was completed in July 2019 and carried out on NCBI PubMed and Web of Science databases according to the Preferred Reporting Items for Systematic Reviews and Meta-Analyses (PRISMA) checklist and flow diagram [41], as shown in Figure 1. We employed the following search strings—(functional near-infrared spectroscopy OR near-infrared spectroscopy) AND Parkinson* OR Multiple Sclerosis OR Mild Cognitive Impairment OR Alzheimer Disease. Two reviewers independently retrieved the articles and assessed their quality, while a third discussant was involved to solve for disagreements concerning the inclusion or exclusion of single articles.

General research strings were intentionally used in order to present a comprehensive scenario of actual applications of this technology. Only full peer-reviewed original articles in English language were included. Each reviewed article is referred to either in a clinical context, rehabilitation program or research study involving at least a group of patients among the selected chronic neurological pathologies and employing fNIRS as one of the main cerebral investigation techniques. Accordingly, case studies and study protocols were excluded. Studies that involved only a group of healthy controls, even if preliminary to pertinent pathological studies, were excluded as well. As a result, we reviewed a total of 63 articles. Among them, 12 concerned applications in PD patients, 11 in MS, 40 in AD continuum represented by this pathology and MCI. Table 1, Table 2 and Table 3 provide an overview of retrieved information in terms of sample size, type of study (i.e., cross-sectional or longitudinal), cortical areas assessed by the fNIRS setup employed, the number of measurement channels (i.e., source-detector pairs) and the tasks to elicit a functional response. In addition, we have also reported the presence of additional neuroimaging modalities and of external devices providing together with fNIRS one or more primary outcomes of the study (i.e., *Multimodal integration* column). Finally, we also kept track of clinical outcomes and neuropsychological measures that were either correlated or used in conjunction with fNIRS-derived hemodynamic variables (i.e., *Integrated clinical outcomes* column). Neuropsychological scales and clinical outcomes were not reported when employed only for the screening and recruitment of patients at group-level.

## 3. fNIRS in Parkinson’s Disease

Parkinson’s Disease (PD) is a slowly progressive neurodegenerative disorder and represents one of the most common pathologies related to ageing, involving about 6.1 million individuals worldwide and presenting considerable implications on national healthcare systems [42,43]. This pathology is characterized by a prominent impairment of motor functions (i.e., bradykinesia, muscular rigidity, rest tremor and postural and gait impairment) together with non-motor features, namely olfactory and autonomic dysfunction, cognitive impairment, psychiatric symptoms, sleep disorders, pain and fatigue [44,45]. As a consequence, complex clinical pictures lead to numerous PD subtypes which can show different clinical profiles and type of progression [46]. We reviewed 12 articles dealing with applications of cerebral fNIRS in a total of 417 PD patients with different disease severity and 202 healthy control (HC) subjects (Table 1). fNIRS was applied either to evaluate Deep Brain Stimulation (DBS) procedures [47,48,49,50] or usual and dual task (DT) walking conditions [51,52,53,54,55,56,57,58]. Most of reviewed articles investigated the cortical activity of the prefrontal cortex (PFC), while only a single study considered the primary motor cortex (PMC) as the region of interest [48]. Finally, one article employed a high-density probe configuration to simultaneously investigate brain activity associated to temporal and occipital cortices [50].

### 3.1. Deep Brain Stimulation

The use of fNIRS and NIRS in the context of DBS is based on the underlying hypothesis that brain stimulation can act on the global cortical neuronal network and that the therapeutic efficacy of this procedure is reflected by the neurovascular activity itself [48]. Sakatani et al. [47] were the first to employ NIRS for assessing changes in PFC oxygenation induced by frequency- and intensity-varying stimulation of the thalamic nucleus ventralis intermedius and globus pallidus internus. They observed HbO2 and HbR changes that were comparable to those induced by cognitive tasks, thus suggesting a possible interaction between the frontal lobes and the deep brain areas stimulated during DBS. More recently, Morishita et al. [48] investigated PMC cortical activity pre-operatively and at 1-month follow-up, while motor scores were assessed by means of the Unified Parkinson’s Disease Rating Scale (UPRS). Results show pre- to post-operative improvement of the UPRS motor score and the group analysis of fNIRS revealed a post-operative cortical activity comparable to the pre-operative one though more confined to the motor cortex for HbO2. Mayer et al. [49] studied the effects of bilateral subthalamic nucleus stimulation on working memory functions. Early PD patients showed a reduced frontal activation with respect to the control group. Overall worsening of working memory performance was accompanied by an increased frontal activation under stimulation and on-medication, while no modifications were observed with respect to medication states, thus suggesting a DBS-induced compensatory mechanism operating within the basal ganglia-prefrontal network. Finally, Eggebrecht et al. [50] utilized a new High-Density Diffuse Optical Tomography (HD-DOT) probe array to map distributed brain functions and resting-state networks in 3 patients undergoing subthalamic nucleus DBS. This study demonstrated that HD-DOT showed a reliable overlap with fMRI, thus suggesting that it can be used to provide individualized functional images when other traditional functional imaging modalities are unavailable.

### 3.2. Walking and Dual Walking Task

Most of reviewed studies in PD were focused on walking and DT conditions to investigate the extent of PFC motor vs. executive and cognitive dysfunction. The underlying hypothesis is that PFC compensates for the motor impairment, hence cortical activation can be considered as an overall index of cognitive load. Within the context of rehabilitation, the promotion of more localized cortical activation associated with executive-attentional functions could be a viable way to activate this compensatory mechanism [54]. Mahoney et al. [51] reported that PD patients showed greater PFC activation in order to successfully achieve the same level of postural stability with respect to age-matched HCs and other mild PD patients. Nieuwhof et al. [52] observed overall increases of HbO2 levels in the PFC during three dual walking tasks (walking while counting forward, serially subtracting, reciting digit spans) compared to a rest condition (standing still). Another study by Cornejo et al. [53] found that both gait stability and PFC activation were enhanced when walking on a treadmill at a self-selected pace with respect to usual walking, suggesting that an external rhythmic pacing may reduce the cognitive mediation on gait. Stuart et al. [54] also found that ageing and pathology affect the PFC compensatory mechanism due to the cognitive control required to perform turning-in-place and walking tasks, while this effect is reduced once the action has begun.

Maidan et al. [55,56,57,58] carried out an extensive work to investigate the relationship between this compensation mechanism and PFC activation and promote strategies to reduce cognitive load. In 2015 the authors studied the interplay of PFC activation during freezing of gait (FOG), a common disturbance among PD patients, associated with anticipated and unanticipated turns, in order to distinguish opposite cognitive requests such as motor planning and reflexive responses [56]. Results revealed that frontal activation levels can be associated to different typologies of FOG. Successively, they studied PD patients without FOG and noticed increased activation during usual walking and a decrease during turning in the absence of cognitive load [55]. Increased activation was also found in a sub-group of patients with impaired ambulation, which further supports the role of PFC in motor-cognitive compensation. Yet another study suggested that a combination of motor and cognitive tasks - namely obstacle negotiation, usual and DT walking—determined different involvements of PFC activity during gait in HCs and PD patients [58]. Finally, the same research group carried out a longitudinal randomized control trial to assess the effects of treadmill training (TT) alone or combined with virtual reality (VR) [57]. PFC activation during obstacle negotiation and DT walking was reduced in the combined TT-VR program condition. However, both experimental conditions induced an overall reduction in the rate of falls post-intervention and an improvement in gait performances, suggesting that simultaneous motor and cognitive training promotes the recruitment of more specific PFC areas.

## 4. fNIRS in Alzheimer’s Disease and Mild Cognitive Impairment

Alzheimer Disease (AD) is a chronic neurodegenerative brain disorder that represents the most common cause of dementia in the elderly [59], affecting about 40–50 million individuals worldwide [60]. Mild Cognitive Impairment (MCI) is defined as a more selective and milder cognitive impairment prior to the onset of AD and its prevalence has been estimated between 3% and 19% in adults older than 65 years-old [61]. The overall number of studies in the literature employing fNIRS as the principal cerebral investigation technique is greater in AD than PD and MS. Moreover, fNIRS appears to have a prognostic role since many studies propose its use to differentiate dementia-related conditions and to identify MCI/AD prodromal stages. Indeed, as reported by several studies, fNIRS allows us to monitor the decline of executive functions and visuospatial abilities, which is an early symptom of these pathologies [62,63]. In most of the reviewed applications, fNIRS probes were placed over the PFC, as this cortical area is mainly associated with high-level processing functions [64]. We identified 40 studies performed with a total of 977 AD and 477 MCI patients (208 amnestic MCI), as well as with 792 age-matched HC subjects (Table 2) Most studies were addressed to the monitoring of tissue oxygenation [65,66,67,68,69,70], the investigation of functional resting-state activity and connectivity [64,71,72,73], the assessment of specific cognitive functions [74,75,76,77,78,79,80,81,82,83,84,85,86,87,88,89,90,91,92,93] such as visuospatial deficits [94,95] and additional ecological applications [96,97]. Though most studies reported cross-sectional reports, we also retrieved six longitudinal ones [98,99,100,101,102,103], thus strongly supporting the idea that fNIRS can represent a viable tool for long-term monitoring of cerebral activity from the early phases of the disease or during rehabilitation.

### 4.1. Tissue Oxygenation Monitoring

More traditional applications of NIRS are those aimed to monitor tissue oxygenation and hemodynamic features such as vasomotor reactivity, cerebral perfusion, autoregulation and metabolism. Indeed, Marmarelis et al. [65] found that NIRS can be used to provide model-based physiological markers of vasomotor and dynamic cerebral autoregulation in amnestic MCI (aMCI) patients with comparable validity to the ones obtained by means of transcranial Doppler (TCD). Viola et al. [66] found a significant correlation between Mini-Mental State Examination (MMSE) scores and reduced tissue oxygenation together with an increased TCD pulsatility index, suggesting this correlation as a prognostic marker of aMCI. Van Beek et al. [67] also noticed from concurrent NIRS-TCD and finger photoplethysmography that the transmission of very low frequency oscillations from cerebral blood flow and systemic blood pressure to changes of HbO2 during repeated sit-stand maneuvers are substantially different between AD patients and HCs. Indeed, AD patients presented an increased transfer function gain and phase lag, which suggest a delay between changes in cerebral blood flow and cortical oxygenation. Likewise, the presence of hypoperfusion, hypometabolism and a non-significant correlation between global brain perfusion and cerebral metabolic rate of oxygen in aMCI patients reported by Liu et al. [68] suggested the presence of an impaired neurovascular coupling. Another study by Babiloni et al. [69] employing NIRS-EEG monitoring under hypercapnia conditions found a reduced vasomotor reactivity and resting-state coherence of aMCI patients compared to controls. In the same line of research, Bär et al. [70] monitored the effects of galantamine treatment on vasomotor reactivity during normocapnia and hypercapnia in AD and vascular dementia patients. Their results also suggested the presence of an altered cerebral autoregulation mechanism in AD, since they found no noticeable decrease of vasomotor reactivity as opposed to vascular dementia patients.

### 4.2. Functional Resting-State

The investigation of functional connectivity and resting-state conditions as monitoring tools of AD progression represents one of the most recent applications. Niu et al. [71] recently explored this information by combining resting-state fNIRS, sliding time window correlation and k-means clustering analysis to reveal an altered dynamic functional connectivity in AD and aMCI patients compared to HC subjects. Zeller et al. [72] found a decrease of low frequency NIRS oscillations in MCI patients and elderly HCs compared to young HCs, while a decrease was observed in MCI patients compared to elderly HCs throughout the parietal cortex. Low frequency oscillations were also correlated with performance on neurophysiological tests. Accordingly, Bu et al. [64] assessed resting-state effective connectivity across the PFC, motor and occipital cortices by means of dynamic Bayesian interference. MCI patients showed a decreased connectivity compared to HCs most notably in the PFC, as well as positive correlations with cognitive performance assessed by means of the MMSE and the Montreal Cognitive Assessment. Significant differences between MCI patients and HCs in inter-hemispheric connectivity during resting-state, as well as letter and category fluency tasks, were also reported by Nguyen et al. [73], thus suggesting that intra- vs. inter-hemispheric connectivity could serve as marker to discriminate MCI from healthy subjects.

### 4.3. Cognitive Tasks–Memory Task

Most of the reviewed studies employed a cognitive task to assess the evoked functional hemodynamic responses in MCI and AD. Among this category, some studies applied a working memory (WM) task, since memory deficits represent a distinctive characteristic of these pathologies. Niu et al. [74] reported that fNIRS may represent a useful tool for the evaluation of cortical functional deficits in cognitive disorders, since decreased frontal and temporal activation was noticed in aMCI during a WM task compared to HCs. In addition, they also found a significant positive correlation between mean HbO2 concentrations and the accuracy rate on the Stroop test. Yeung et al. [75] suggested that in MCI the performance on the WMT is altered by an impairment at the PFC level because the measured activation did not depend on the cognitive load of the employed n-back task. A lower WM task performance associated with a decreased bilateral dorsolateral PFC activation was also reported by Uemura et al. [86], who proposed this finding as a predictive marker of cognitive decline. Likewise, region-specific variations (i.e., frontal, dorsolateral PFC and parietal cortex) of HbO2 levels in AD patients were observed during a single-word presentation task by Kato et al. [87], while Ateş et al. [88] were able to discriminate AD from HCs based on reaction times and ventral PFC activity during an emotional WMT. Oboshi et al. [89] found a significant correlation between task-evoked cerebral blood flow in the PFC and α4β2 nicotinic acetylcholine receptor availability assessed by PET during a visual WM task in both AD patients and HCs. However, the WM task performance and the averaged response were lower and delayed in the AD group compared to HCs, suggesting that the α4β2 nicotinic acetylcholine receptor system supports PFC activity associated with cognitive tasks. Recently, Li et al. [91] investigated the hemodynamic response patterns of MCI and AD patients of different severity during a verbal digit span task. Lower HbO2 concentrations across frontal and bilateral parietal cortices were associated with the severity of the disease, while MMSE scores were significantly correlated with all fNIRS-derived hemodynamic indices. A successive concurrent fNIRS-EEG study investigated AD-related alterations in cortical network during a digit verbal span task by means of connectivity analysis and graph-based indices [90], revealing the presence of region-specific and EEG frequency-linked variations compared to HCs. Specifically, AD patients presented a weaker connectivity in high alpha and beta bands related to orbitofrontal and parietal regions, while at the frontal and medial orbitofrontal regions a lower degree and clustering coefficient was found. Recent studies also compared fNIRS with standard clinical outcome measures for diagnostic and screening purposes in AD patients. Accordingly, Perpetuini et al. [92,93] suggested that fNIRS can be used as a flexible neuroimaging technology for the administration of free and cued selective reminding test and for the assessment of visuospatial and short-term memory abilities. A multivariate analysis of task-evoked activity provided acceptable specificity and sensitivity to AD diagnosis and comparable test performances to the predictive value of the clock drawing test [92].

### 4.4. Cognitive Tasks–Verbal Fluency Task

Several studies used fNIRS during the administration of verbal fluency tasks (VFT), namely category and letter fluency, which have been demonstrated to be reliable indicators of AD-related neurodegeneration, frontal lobe dysfunction and lexical-semantic vs. executive-control impairment that conversely characterize traumatic brain injury [104]. Hock et al. [76,77] reported and successively confirmed in a simultaneous NIRS-PET study the reduction of bilateral PFC and parietal cortical activation in AD patients compared to HCs. This effect was also dependent on the age of participants. Additionally, significant positive correlations between regional cerebral blood flow measured by PET and total oxygenation changes during the Stroop color word interference test were found. In addition, Fallgatter et al. [78] reported significant differences of hemispheric activation (i.e., lateralization) between AD and HCs associated to letter and category VFTs. AD patients presented a lower VFT performance associated to a lack of lateralization conversely shown in HCs. These findings were successively extended by means of a multi-channel fNIRS system by Herrmann et al. [79] who also found that AD patients show reduced HbO2 levels and a more distributed pattern of cortical activation than HCs. Arai et al. [80] also reported a significant decrease of frontal and bilateral parietal activation in AD patients, while this effect was observed only in the right parietal cortex in MCI. However, a more recent study by Yeung et al. [81] showed that a cohort of MCI patients had no lateralized response during the category fluency task compared to HCs, suggesting that the absence of lateralization may indicate a cortical reorganization following PFC impairment. Yap et al. [82] also suggested an impaired compensatory mechanism during semantic VFT in mild AD and MCI patients due to cortical hypo- and hyper-activation respectively. Moreover, Doi et al. [83] reported increased PFC activity in MCI during DT walking compared to usual walking and a correlation with executive functions assessed by a modified Stroop color-word interference task. Findings by Katzorke et al. [84] also stressed the importance of investigating the influence of the temporal cortex in VFTs as they reported a lower activation of inferior frontotemporal cortex in MCI compared to HCs. Finally, fNIRS paired with VFT can also be used to assess dementia-related impairments in patients with diagnoses other than MCI and AD. For instance, Metzger et al. [85] reported that patients with the behavioral-subtype of frontotemporal dementia show a significantly different cortical activation pattern during letter and category fluency tasks compared to HCs and AD patients.

### 4.5. Cognitive Tasks–Visuospatial Task

Few fNIRS applications aimed to study task-evoked cortical activity associated with visuospatial tasks. Since depressive features in the aged population are often misinterpreted as initial stages of AD, Kito et al. [94] compared brain activity of AD patients, age-matched HCs and patients with late life depression. As a result, people with late life depression showed a significant decrease of parietal cortex activation compared to AD patients during a visuospatial task. On the other hand, Zeller et al. [95] assessed the correlates of visuospatial deficits in AD by employing the modified version of the Benton Line Orientation task. Their results confirmed that fNIRS could be a viable tool for early detection of AD and the assessment of disease progression.

### 4.6. Ecological Applications

Two additional studies were dedicated to ecological applications of fNIRS in the context of more specific research questions. Tomioka et al. [96] used a driving simulator during a collision avoidance task in order to test the timing of decision-making in a cohort of AD patients. Decreased PFC activation and a negative correlation with the delay in braking suggested that rapid decision-making is impaired in these patients. Conversely, Shimizu et al. [97] assessed the efficacy of movement music therapy in MCI patients during a 3-month randomized, controlled, single-blinded intervention trial. Differently from the control group, the intervention group experienced significant increase in both PFC activity and physical functional tests, therefore confirming that stimulating physical and cognitive functions may reduce the decline associated to the pathology.

### 4.7. Longitudinal Monitoring

Finally, a series of longitudinal studies employed fNIRS as one of the primary outcome measures, hence reinforcing the idea that previous cross-sectional applications can be applied for studying changes over time. Van Beek et al. [98] showed that AD patients treated with the cholinesterase inhibitor galantamine exhibited a greater reduction of HbO2 and total hemoglobin concentration levels compared to controls during hypotension induced by a single sit-stand task, suggesting an increased ischemic vulnerability due to impaired cortical perfusion. In a 24-week study, Araki et al. [101] explored the combined effects of memantine and donepezil on behavioral, psychological and cognitive functions in a group of AD patients. NIRS measurements and the accuracy on the letter fluency task were also significantly correlated with primary outcome measures, namely the MMSE score, Clock Drawing Test and the Clinical Global Impression-Improvement scale. In addition, Metzger et al. [102] observed an overall improvement on letter and category fluency tasks, as well as on PFC and speech areas activation that supported the positive neurobiological effects of cholinesterase inhibition on cognitive functions in AD patients. Concerning other applications not involving the administration of medications, Viola et al. [99] showed in a prospective, controlled and open-labeled study that a rehabilitation therapy of mild AD patients based on hypercapnia improved both frontal oxygenation and performances on the MMSE and Rey Auditory Verbal Learning Test scores. Vermeij et al. [100] showed that 5-weeks of adaptive computerized WM training improved the behavioral performance of MCI patients on a verbal n-back task at low cognitive load. At a high cognitive load, more evident hemodynamic responses were associated with higher training gains, thus indicating that PFC activity may be considered as predictive of WM training effectiveness. Finally, Polak et al. [103] are currently conducting a 10-year study aimed at evaluating, whether vagus somatosensory evoked potentials and fNIRS, together with several other outcome measures (Table 2), could serve as future diagnostic markers of AD and as a monitoring tool of cognitive decline. This study is expected to end in 2021 but several results have already been published, such as the work by Zeller et al. [72] and Katzorke et al. [84] already mentioned in this review.

## 5. fNIRS in Multiple Sclerosis

Multiple Sclerosis (MS) is a chronic immune-mediated disease of the central nervous system with different clinical courses and onset in early adulthood [105]. It has been estimated that 2–3 million people are affected by MS worldwide, while the prevalence is 50–300 per 100,000 individuals [106]. Disability buildup due to the progression of neuropathology heavily affects the quality of life of affected persons, requiring personalized rehabilitation strategies to support motor and cognitive functions in daily life [107,108]. We reviewed 11 studies concerning applications of cerebral fNIRS with a total of 242 MS patients with either primary progressive (PPMS), secondary progressive (SPMS) or relapsing-remitting (RRMS) disease courses, as well as 128 HC subjects (Table 3). Most studies investigated cortical activity in areas of the frontal, with the exception of two studies employing fNIRS probes placed over motor areas [109,110]. All studies were cross-sectional and employed different tasks, ranging from simple DT [111], static motor protocols [110,112] and WM tasks [113] to combined walking and other DT conditions [109,114,115,116]. We also found applications of NIRS to assess ozone autohemotherapy [117,118] and the extent of microvascular hemoglobin saturation [119].

Borragàn et al. [111] employed a simultaneous WM and parity number decision paradigm to evaluate the effects of high and low cognitive fatigue and sleep episodes in RRMS patients. By adapting cognitive demand according to individual maximal performance, a significant relationship between the perceived cognitive fatigue and dorsolateral PFC activation was found. Furthermore, the triggering of cognitive fatigue during the task was associated with an increased number of hours slept. Stojanovic-Radic et al. [113] reported that MS patients show an increased or decreased left superior frontal gyrus activation depending on low or high difficulty levels of the n-back WM task, while the opposite trend was evidenced in HCs. In accordance with previous fMRI findings, performance accuracy at the low difficulty level was significantly reduced in MS suggesting that different brain areas are progressively recruited to cope with neural damage. Jimenez et al. [110] and Wolff et al. [112] applied non-walking motor protocols to assess task-evoked cortical activity. In the former study, MS patients exhibited significantly lower activation and interhemispheric communication than HCs during finger tapping but not during a resting condition [110]. Conversely, Wolff et al. [112] recently evaluated the relationship between motor-cognitive fatigue and self-control during a strenuous physical task and found a significant increase of both perceived motor-cognitive exertion and PFC activation across trials.

Moving to studies that used a combination of walking and DT protocols, Chaparro et al. [115] observed that partial body weight support while walking allowed to maintain the same activation levels across DT conditions of increasing complexity. This result would suggest a potential effect of rehabilitation over PFC activity and spatiotemporal features of gait. Saleh et al. [109] assessed the effects of DT on walking and cognitive performances on bilateral premotor and supplementary motor areas. MS patients showed significant DT effects on cognitive performance, while no differences were found for gait with respect to age-matched HCs. Results also suggest that the role of right premotor cortex is altered in MS, since no differences in cortical activation during DT compared to usual walking were found, while HCs exhibited increased activation. Hernandez et al. [114] reported that disability levels affect PFC activation during locomotion. Indeed, while no significant differences from HCs were found for walking performance, MS patients showed significant increases of HbO2 levels, especially during DT compared to usual walking. In a subsequent study, the same authors investigated the effects of a concurrent cognitive task on balance and coordination. Results showed that MS affects the ability to recruit additional cognitive resources under DT conditions and that several spatiotemporal gait characteristics are strongly associated with mean HbO2 levels [116]. These authors also suggested that future rehabilitation programs may benefit of interventions focused on improving attention and cognitive functions during walking with interference tasks. At the same time, serial fNIRS studies throughout rehabilitation may be useful to monitor therapeutic effectiveness.

In remaining studies, cortical activity of MS patients was not assessed by means of motor or cognitive tasks. Yang and Dunn [119] monitored cortical hemoglobin saturation (StO2) over the bilateral frontal cortex to evaluate whether this technique can serve as potential marker of hypoxia in MS. Forty two percent of 72 patients showed a reduction of StO2 greater than 2 standard deviations from the mean value of HCs, as well as the lack of dependence on brain-to-scalp distance and a significant correlation with clinical measures of motor disability, disease duration and Expanded Disability Status Scale. Finally, Molinari et al. [117,118] employed NIRS to monitor tissue oxygenation throughout the frontal cortex during ozone autohemotherapy in RRMS patients, showing that NIRS can provide a reliable and non-invasive means to characterize treatment effects on cerebral hemodynamic responses. In a previous study, they evaluated the long-term effects of ozone autohemotherapy on cerebral metabolism and vasomotor reactivity, finding increased oxygenation at the end of treatment that allowed them to distinguish patients from HCs based on hemodynamic-derived variables and Cytochrome-c-oxidase (CYT-c) activity [117]. Analogous results were also found in a replication study, where they assessed the effects of ozone autohemotherapy without monitoring CYT-c and extended previous considerations using both linear and non-linear analysis methods [118].

## 6. Discussion

In this systematic review, we have provided a comprehensive scenario of existing peer-reviewed fNIRS and NIRS studies that investigated several functional aspects of the brain in the most common chronic neurological conditions. Other review articles dedicated to fNIRS applications in ageing individuals and neurological disorders are present in the literature. Among them, Mihara and Miyai [26] reported the applications of fNIRS as either a monitoring or a therapeutic tool in neurorehabilitation, focusing on functional reorganization after brain damage, motor learning, brain-computer interface (BCI) and neurofeedback training. Arenth et al. [120] introduced the potential advantages of this technique in neurorehabilitation, motor, visual and cognitive tasks, even if a limited number of studies on neurological populations were present at that time. Another review on fNIRS applications in patients affected by gait and movement disorders (i.e., stroke and PD) is provided by Gramigna et al. [28], whereas Herold et al. [121] examined the combined effects of physical exercise and cognitive testing and Naseer and Hong [122] further investigated fNIRS-BCI and related processing methods. To the best of our knowledge, however, none of the above reviews provided an overall scenario of fNIRS applications directly addressing chronic neurological conditions. The only neurological condition that has been extensively investigated with fNIRS is stroke, which has been recently reviewed by Yang et al. [35].

We have therefore focused our attention on the current state-of-the-art of fNIRS applications dealing with motor and cognitive disability in the most frequent chronic neurological disorders of the elderly, namely PD and the AD continuum and MS as the most frequent chronic neurological disease of young and middle-aged adults. Our results reveal that AD and MCI are the most frequently addressed conditions by a wide variety of fNIRS applications, ranging from the monitoring of tissue oxygenation and vasomotor reactivity [65,66,67,68,69,70], to the analysis of resting-state conditions [64,71,72,73], the assessment of cognitive functions by means of memory and verbal fluency tasks [74,75,76,77,78,79,80,81,82,83,84,85,86,87,88,89,90,91,92,93], visuospatial functions [94,95] and other ecological applications [96,97]. Within the broad context of PD and MS, reviewed studies were more focused on the assessment of motor functions [51,53,54,55,56,109,110,112]. In some cases, walking and other motor protocols were also assessed in dual-task conditions [52,57,58,114,115,116]. Finally, we also found a limited number of applications where fNIRS or NIRS was used to assess specific treatment strategies, such as ozone autohemotherapy in MS [117,118] and DBS in PD patients [47,48,49,50].

Most of the reviewed articles employed fNIRS as a surrogate biomarker to characterize clinical phenotypes rather than to devise targeted interventions to promote neuroplasticity. In particular, studies in MS and PD patients focused on motor-related features of these neurological disorders, while studies of the AD continuum were mainly addressing neurocognitive domains of executive functions, learning and memory as stated in the fifth edition of Diagnostic and Statistical Manual of Mental Disorders [123]. Within the above clinical conditions, the scarcity of intervention-based applications represents a partial limitation to potential fNIRS capabilities, since this technology can be already employed to monitor functional changes and neuroplasticity mechanisms occurring at the cortical level. Neuroplasticity is sustained by complex mechanisms that are region-specific, depend on brain development and are substantially different between healthy and disease conditions [124]. Moreover, several studies have validated cortical activation measured with fNIRS by comparing it to the fMRI-BOLD signal during sensory-motor and cognitive tasks, as well as during resting-state, confirming that fNIRS is able to provide a valid, albeit indirect, measure of neural activity [19]. Consequently, even if fNIRS does not substitute fMRI, this technique will possibly provide several benefits for both clinical practice and research [125] thanks to its potential ability to serve as diagnostic and monitoring tool.

The advantages of non-invasiveness, portability and ability to provide longitudinal measurements favor a high patient compliance and render fNIRS a technology suitable for carrying out multiple measurements in rehabilitation settings. Two observations can be done with respect to the potential advantages offered by this technique. First, from Table 1, Table 2 and Table 3 it emerges that only 9 out of 63 reviewed studies applied a longitudinal approach. Nonetheless, the remaining cross-sectional studies provided relevant methodological considerations that can be usefully translated into longitudinal applications to monitor the progression of chronic diseases. In turn, long-term monitoring of neuroplasticity would serve as a reference to implement individualized intervention strategies. Moreover, fNIRS probes can be placed over cerebral areas known to be involved by the employed task, hence allowing personalization of experimental protocols. Several studies employed a reduced number of measurement channels that were placed over the PFC, since it is reported to play an important role in age-related compensatory mechanisms and in neurofunctional changes according to the CRUNCH model [126]. In particular, this model states that PFC activity induced by cognitive tasks favors the recruitment of additional cerebral areas, also in the ipsilateral hemisphere, serving as an age-related compensatory mechanism. Though a limited number of measurement channels does not actually represent an absolute limitation, in the context of chronic neurodegenerative diseases it seems more advisable to employ more extended probe configurations to simultaneously map task-evoked hemodynamic responses over heterogeneous cortical areas and to allow multimodal imaging as well.

The actual technical limitations restricting fNIRS usage for both research and clinical purposes are a lower spatial resolution with respect to other neuroimaging technologies, such as fMRI and a higher contamination of the signal by systemic and extracerebral sources [32,127]. Indeed, lateral resolution is limited by the length of measurement channels over the scalp, which also affects the depth resolution of the measurements [34]. A possible solution to this problem, at the expense of portability, are HD-DOT systems, which employ overlapping source-detector measurements at multiple separation distances for mapping cortical activity over an extended field of view with a spatial resolution comparable to that of fMRI [20,128]. Limits to the spatial resolution can be also overcome by integrating measured HbO2 and HbR signals with patient- or atlas-based structural MRI data and to obtain a better visualization and interpretation of results [129,130]. On the other hand, a still unresolved issue that highly impacts both reproducibility and interpretation of results is the lack of standardization of pre-processing and analysis pipelines [30,33]. This issue can be attributed to the differences in available systems and the heterogeneity of methodological approaches and research objectives, hence demanding specific considerations over single sub-steps of data processing pipelines [32]. At the moment, preliminary attempts to develop standardized pipelines involve mainly methodological aspects of data pre-processing and analysis [131] rather than commonly accepted guidelines, while standard requirements to produce fNIRS devices have been published in 2015 (*IEC 80601-2-71:2015 Medical electrical equipment–Part 2-71—Particular requirements for the basic safety and essential performance of functional near-infrared spectroscopy (NIRS) equipment*).

Among reviewed studies, few of them assessed the association between fNIRS-derived hemodynamic variables with standard clinical outcomes and/or scores on neuropsychological tests, since most of these measures were used for patient characterization at a group-level (i.e., definition of the overall severity of impairment or statement of recruitment criteria). However, limiting the analysis of hemodynamic responses to the cross-sectional investigation of cortical activation patterns would not provide insight concerning the effects of therapeutic interventions within the theoretical framework of neuroplasticity. Indeed, in a longitudinal setting it would be desirable to correlate cortical fNIRS activation patterns to treatment outcomes in order to assess whether the former are associated to successful or unsuccessful interventions and, consequently, if hemodynamic changes reflect either an adaptive or a maladaptive neuroplasticity [132]. In turn, this approach would possibly provide a better definition of experimental designs and translational research questions as a future perspective. Furthermore, research lines and clinical applications of fNIRS would benefit from the multimodal integration with other neuroimaging methods in order to provide a more comprehensive anatomical and functional representation of patients’ status. In addition to the merging of functional and anatomical information by fNIRS and MRI, photoacoustic methods allows the mapping of functional activity beyond the optical resolution limit due to scattering and absorption of biological tissues, showing promising results for the real-time and high-resolution investigation of deep brain activity in neuroimaging applications over small animals [133,134,135]. As well, photoacoustic holds major promises for the simultaneous assessment of stimulus-evoked hemodynamic responses, calcium and voltage dynamics in brain tissues [136,137,138,139], therefore supporting fNIRS in the translational research of a more portable investigation of neurological disorders and the mechanisms of neurovascular coupling in general.

## 7. Conclusions

In conclusion, the available literature supports the idea that fNIRS can be a viable tool to detect functional differences between normal ageing and people affected by the most common chronic neurological disorders, namely PD, MS and the AD continuum. We found that this technology is mainly employed for the characterization of the patients’ clinical phenotype, whereas a systematic adoption of intervention-based monitoring still remains to be seen. Overall results open the scenario to the possibility of employing this low-cost and portable technology as a monitoring tool of cerebral plasticity during disease progression and to promote subject-specific intervention strategies.

## Figures and Tables

**Figure 1 diagnostics-10-00581-f001:**
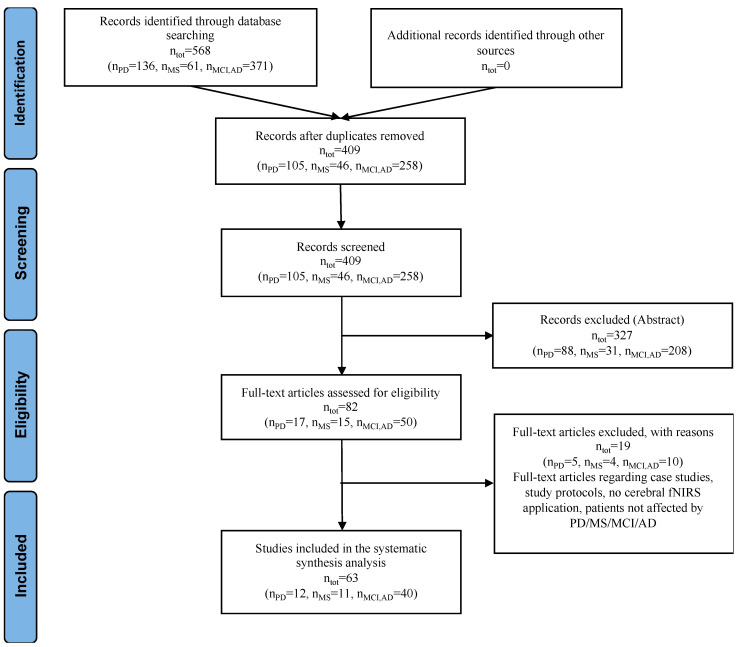
Preferred Reporting Items for Systematic Reviews and Meta-Analyses (PRISMA) checklist and flow diagram with information about the search, screening and selection processes performed in order to identify the relevant articles included in this review.

**Table 1 diagnostics-10-00581-t001:** Summary of peer-reviewed articles of functional Near-Infrared Spectroscopy (fNIRS) applications in Parkinson’s Disease (PD) patients found trough literature search of NCBI PubMed and Web of Science databases.

Parkinson’s Disease (PD)									
		fNIRS Device	Patient Info	Study Type	Integrated Clinical Outcomes	Cortical Areas	Performed Task	Ch.	Multimodal Integration
**Deep Brain Stimulation**	**Sakatani et al., 1999**	NIRO-300 (Hamamatsu Photonics K.K., Japan)	5 PD, 1 essential tremor patient	cross-sectional	UPDRS, tremor rating scale	bilateral PFC	tissue oxygenation monitor	1	no
	**Morishita et al., 2016**	FOIRE-3000 (Shimadzu Corporation, Kyoto, Japan)	6 PD	longitudinal	UPDRS	primary motor cortex	unilateral hand movement	48	no
	**Mayer et al., 2016**	ETG-4000 (Hitachi Medical Co., Tokyo, Japan)	9 PD, 8 HC	cross-sectional (*)	UPDRS	lateral and medial FC (bilateral hemispheres)	spatial-delayed response task (working memory task)	22	no
	**Eggebrecht et al., 2014**	custom HD-DOT system	(18 HC) 3 PD	cross-sectional	n.d.	almost whole-head (temporal, occipital cortex)	auditory words, resting-state	1200+	T1- and T2-MRI, fMRI (not performed on PD patients)
**Walking and Dual Walking Task**	**Mahoney et al., 2016**	fNIR Imager 1000 (fNIR Devices LLC, Protomac, MD, USA)	26 PD, 117 mild PD, 126 HC	cross-sectional	n.d.	PFC	postural stability control task	16	instrumented walkway
	**Nieuwhof et al., 2016**	PortLite (Artinis Medical Systems, Elst, The Netherlands)	14 PD	cross-sectional	n.d.	bilateral PFC	counting forward, serially subtracting, reciting digit spans	3	instrumented walkway with pressure sensors
	**Cornejo et al., 2018**	PortLite (Artinis Medical Systems, The Netherlands)	20 PD	cross-sectional	n.d.	dorsolateral PFC and anterior PFC (bilateral BA10)	over-ground and treadmill walking	3	instrumented treadmill, 3D-accelerometers
	**Stuart et al., 2019**	Oxymon (Artinis Medical Systems, The Netherlands)	24 PD, 19 HOA, 25 HYA	cross-sectional	n.d.	PFC (BA9 and BA10)	turning-in-place task	n.d.	no
	**Maidan et al., 2015**	Oxymon MKIII (Artinis Medical Systems, The Netherlands)	11 PD, 11 HC	cross-sectional	n.d.	PFC (bilateral BA10)	walking with anticipated and unanticipated turns (WT)	6	no
	**Maidan et al., 2016**	PortLite (Artinis Medical Systems, Elst, The Netherlands)	68 PD, 38 HC	cross-sectional	n.d.	dorsolateral PFC and anterior PFC (bilateral BA10)	obstacle negotiation, WT, DWT	3	instrumented walkway with pressure sensors
	**Maidan et al., 2017**	PortLite (Artinis Medical Systems, Elst, The Netherlands)	49 PD	cross-sectional	n.d.	dorsolateral PFC and anterior PFC (bilateral BA10)	usual walking and turning (WT)	3	instrumented walkway with pressure sensors
	**Maidan et al., 2018**	PortLite (Artinis Medical Systems, Elst, The Netherlands)	64 PD	longitudinal	UPDRS	dorsolateral PFC and anterior PFC (bilateral BA10)	treadmill training (obstacle negotiation, WT, DWT)	3	instrumented walkway with pressure sensors

DWT dual walking task; FC frontal cortex; HC healthy controls; HOA healthy old adults; HYA healthy young adults; PFC prefrontal cortex; UPDRS Unified Parkinson’s Disease Rating Scale; WT walking task; (*) cross-sectional study with multiple fNIRS acquisitions.

**Table 2 diagnostics-10-00581-t002:** Summary of peer-reviewed articles of fNIRS applications in AD and Mild Cognitive Impairment (MCI) patients found trough literature search of NCBI PubMed and Web of Science databases.

		Mild Cognitive Impairment (MCI) and Alzheimer’s Disease (AD)							
		fNIRS Device	Patient Info	Study Type	Integrated Clinical Outcomes	Cortical Areas	Performed Task	Ch.	Multimodal Integration
**Tissue Oxygenation Monitoring**	**Marmarelis et al., 2017**	N.D. (Hamamatsu Photonics K.K., Japan)	(46) 38 aMCI, (22) 14 HC	cross-sectional	n.d.	PFC	tissue oxygenation monitor	n.d.	TCD, finger photo-plethysmography, capnography
	**Viola et al., 2013**	custom NIRS device (T-NIRS EVO II)	21 aMCI, 10 HC	cross-sectional	MMSE	bilateral frontal and parieto-temporal cortex	tissue oxygenation monitor	1	TCD
	**van Beek et al., 2012**	Oxymon (Artinis Medical Systems, Zetten, The Netherlands)	21 mild to moderate AD, 20 HC	cross-sectional	n.d.	bilateral FC	tissue oxygenation monitor during repeated sit-stand manoeuvres	n.d.	TCD, finger photo-plethysmography, ECG
	**Liu et al., 2014**	NIRO-200NX (Hamamatsu Photonics K.K., Japan)	32 aMCI, 21 HC	cross-sectional (**)	n.d.	n.d.	tissue oxygenation monitor	2	color-coded duplex ultrasonography, MRI, sphygmomanometer, pulse oximeter, ECG, capnography
	**Babiloni et al., 2014**	ISS oximeter, Model 96,208 (ISS Inc., Champaign, IL, USA)	10 aMCI, 10 HC	cross-sectional	MMSE	bilateral PFC	tissue oxygenation monitor under resting-state and hypercapnia conditions	2	concurrent NIRS-EEG
	**Bär et al., 2007**	NIRO-500 (Hamamatsu Photonics K.K., Japan)	17 AD, 17 vascular dementia patients, 20 HOA, 20 HYA	cross-sectional	MMSE	left FC	tissue oxygenation monitor under normocapnia and hypercapnia conditions	2	TCD, finger blood pressure monitor
**Functional Resting-State**	**Niu et al., 2019**	CW6 (TechEn Inc., Milford, MA, USA)	23 AD, 25 aMCI, 30 HC	cross-sectional	MMSE, AVLT, MoCA	whole-head	resting-state	46	no
	**Zeller et al., 2019**	ETG-4000 (Hitachi Medical Co., Tokyo, Japan)	54 MCI, 61 HOA, 25 HYA	cross-sectional	DemTect, RCFT, RWT, VLMT, TAP, WMS-R	frontal and parietal cortex	resting-state	96	no
	**Bu et al., 2019**	NirSmart (Danyang Huichuang Medical Equipment Co., PR China)	26 MCI, 28 HC	cross-sectional	MMSE, MoCA	bilateral PFC, motor and occipital cortex	resting-state	14	no
	**Nguyen et al., 2019**	Custom fNIRS device	42 MCI, 42 HC	cross-sectional	MMSE	PFC	resting-state, oddball task, 1-back task, letter and category fluency task	4	no
**Cognitive Task - Memory Task**	**Niu et al., 2013**	ETG-4000 (Hitachi Medical Co., Tokyo, Japan)	8 aMCI, 16 HC	cross-sectional	MMSE, AVLT, BNT, Stroop Test	bilateral frontal, parietal and temporal cortex	n-back task (WMT)	54	no
	**Yeung et al., 2016**	OEG-SpO2 system (Spectratech Inc., Tokyo, Japan)	10 aMCI, 16 MCI, 26 HC	cross-sectional	n.d.	bilateral PFC	n-back task	16	no
	**Uemura et al., 2016**	FOIRE-3000 (Shimadzu Corporation, Kyoto, Japan)	64 aMCI, 66 HC	cross-sectional	n.d.	bilateral prefrontal and frontopolar cortex (BA9, 46, 10)	memory encoding and delayed retrieval task	22	no
	**Kato et al., 2017**	ETG-4000 (Hitachi Medical Co., Tokyo, Japan)	42 AD, 98 intermediate group (65 high score HDS-R/MMSE, 33 low score HDS-R/MMSE), 91 HC	cross-sectional	MMSE, HDS-R, Z-score of VSRAD	FC, dorsolateral PFC, bilateral parietal cortex	single-word presentation task	44	MRI
	**Ateş et al., 2017**	ETG-4000 (Hitachi Medical Co., Tokyo, Japan)	20 AD, 20 HC	cross-sectional	n.d.	dorsolateral and ventral PFC	emotional working memory (n-back task)	24	no
	**Oboshi et al., 2016**	OEG-16 system (Spectratech Inc., Yokohama, Japan)	11 early-to-moderate AD, 11 HC (*)	cross-sectional	alfa4beta2 nicotinic receptor tracer [18F]2FA	PFC	visual WMT	16	PET
	**Li et al., 2018**	NIRScout (NIRx Medizintechnik GmbH, Germany)	9 MCI, 13 AD (6 mild AD, 7 moderate to severe AD), 8 HC	cross-sectional	MMSE	FC, bilateral parietal cortex	digit verbal span task	46	no
	**Li et al., 2019**	NIRScout (NIRx Medizintechnik GmbH, Germany)	14 mild AD, 8 HC	cross-sectional	n.d.	FC, bilateral parietal cortex	digit verbal span task	46	concurrent fNIRS-EEG
	**Perpetuini et al., 2017**	Imagent (ISS Inc., Champaign, IL, USA)	11 mild AD, 11 HC	cross-sectional	free and cued selective reminding test	bilateral PFC	free and cued selective reminding test	17	no
	**Perpetuini et al., 2019**	Imagent (ISS Inc., Champaign, IL, USA)	11 mild AD, 11 HC	cross-sectional	CDT	bilateral FC and PFC	CDT, digit span test, Corsi block tapping test	21	no
**Cognitive Task - Verbal Fluency Task (VFT)**	**Hock et al., 1996**	NIRO-500 (Hamamatsu Photonics K.K., Japan)	(17 HOA, 12 HYA) 19 AD, 19 HC	cross-sectional	n.d.	bilateral PFC and parietal cortex	calculation task	2	no
	**Hock et al., 1997**	NIRO-500 (Hamamatsu Photonics K.K., Japan)	29 mild AD, 27 HC	cross-sectional	n.d.	frontal, prefrontal and parietal cortex	letter fluency, modified Stroop colour word interference test	2	concurrent NIRS-PET
	**Fallgatter et al., 1997**	Critikon 2020 Cerebral Redox Monitor (Johnson and Johnson Medical)	10 AD, 10HC	cross-sectional	n.d.	bilateral PFC	letter and category fluency tasks	4	no
	**Herrmann et al., 2008**	ETG-100 (Hitachi Medical Co., Tokyo, Japan)	16 AD, 16 HC	cross-sectional	n.d.	bilateral PFC	letter and category fluency tasks	24	no
	**Arai et al., 2006**	ETG-7000 (Hitachi Medical Co., Tokyo, Japan)	15 AD, 15 MCI, 32 HC	cross-sectional	MMSE	FC, occipital cortex, bilateral parietal cortex	letter fluency task	84	no
	**Yeung et al., 2016**	OEG-SpO2 system (Spectratech Inc., Tokyo, Japan)	10 aMCI, 16 MCI, 26 HC	cross-sectional	BNT, HKLLT, RCFT, STT	bilateral PFC	category fluency task	16	no
	**Yap et al., 2017**	OT-R40 fNIRS topography system (Hitachi Medical Co., Tokyo, Japan)	18 mild AD, 12 MCI, 31 HC	cross-sectional	MMSE	PFC, partial temporal cortex	semantic fluency task	52	no
	**Doi et al., 2013**	OEG-16 system (Spectratech Inc., Yokohama, Japan)	16 MCI	cross-sectional	modified Stroop colour and word test	bilateral PFC	normal and dual-task walking (letter fluency task)	16	no
	**Katzorke et al., 2018**	ETG-4000 (Hitachi Medical Co., Tokyo, Japan)	55 MCI, 55 HC	cross-sectional	n.d.	FC, PFC and temporal cortex	letter and category fluency task, control condition	52	no
	**Metzger et al., 2016**	ETG-4000 (Hitachi Medical Co., Tokyo, Japan)	8 bvFTD, 8 AD, 8 HC	cross-sectional	n.d.	bilateral PFC and temporal cortex	letter and category fluency task, control condition	22	no
**Visuospatial Task**	**Kito et al., 2014**	FOIRE-3000 (Shimadzu Corporation, Kyoto, Japan)	30 patients with depression, 28 AD, 33 HC	cross-sectional	MMSE, CDR, FAB, HAMD	FC and parietal cortex	letter fluency task, Benton Judgment of Line Orientation	44	no
	**Zeller et al., 2010**	ETG-100 (Hitachi Medical Co., Tokyo, Japan)	13 mild to moderate AD, 13 HC	cross-sectional	n.d.	parietal cortex	modified version of the Benton Line Orientation Task	24	no
**Ecological Applications**	**Tomioka et al., 2009**	ETG-4000 (Hitachi Medical Co., Tokyo, Japan)	12 AD, 14 HC	cross-sectional	n.d.	bilateral PFC and temporal cortex	collision avoidance (driving task)	52	driving simulator
	**Shimizu et al., 2018**	LABNIRS (Shimadzu Corporation, Kyoto, Japan)	45 MCI (35 intervention group, 10 control group)	longitudinal	FAB, CS-30 test, one-leg standing test, sit-and-reach test, timed Up & Go test	bilateral PFC	movement music therapy (physical-cognitive task)	45	digital sit-and-reach instrument box, digital handgrip dynamometer, walking measurement instrument
**Longitudinal Applications**	**van Beek et al., 2010**	Oxymon (Artinis Medical Systems, The Netherlands)	21 AD, 20 HC	longitudinal	n.d.	bilateral FC	tissue oxygenation monitor during postural change task	n.d.	TCD, pulse-oximeter, capnography, finger photo-plethysmography
	**Araki et al., 2014**	ETG-4000 (Hitachi Medical Co., Tokyo, Japan)	37 moderate-to-severe AD (19 experimental group, 18 control group)	longitudinal	MMSE, CDT, CGI-I scale, NPI, J-ZBI	FC	letter fluency task	22	no
	**Metzger et. al, 2015**	ETG-4000 (Hitachi Medical Co., Tokyo, Japan)	24 AD	longitudinal	MMSE, immediate and delayed word list recall trials	bilateral PFC and temporal cortex	letter and category fluency task	44	no
	**Viola et al., 2014**	custom NIRS instrument (T-NIRS EVO II)	25 mild AD	longitudinal	MMSE, AVLT	bilateral frontal and parieto-temporal cortex	tissue oxygenation monitor	1	no
	**Vermeij et al., 2017**	Oxymon MKIII (Artinis Medical Systems, The Netherlands,)	14 MCI, 21 HC	longitudinal	n.d.	bilateral PFC	verbal n-back task	n.d.	finger photoplethysmography, ECG
	**Polak et al., 2017s**	ETG-4000 and ETG-100 (Hitachi Medical Co., Tokyo, Japan)	530 AD, 74 MCI	longitudinal	MMSE, Anxiety Status Inventory, Bayer Activities of Daily Living Scale, BDI-II, DemTect, Edinburgh test of handedness, GDS, HAMD, RCFT, RWT, VLMT, TAP, WMS-R	prefrontal and parietal cortex	resting-state, letter and category fluency task, trail making test, angle discrimination test	52 and 24	blood test, vagus somatosensory evoked potentials (EEG), intima media thickness, left ventricular ejection fraction, MRI, PET, CSF analysis

aMCI amnestic mild cognitive impairment; AVLT Auditory Verbal Learning Test; BDI-II Beck’s Depression Inventory-II; BNT Boston Naming Test; bvFTD behavioral-subtype frontodementia; CS-30 30 s Chair Stand Test; CDT Clock Drawing Test; CDR Clinical Dementia Rating; CGI-I Clinical Global Impression-Improvement scale; FAB Frontal Assessment Battery; FC frontal cortex; GDS Geriatric Depression Scale; HAMD Hamilton Rating Scale for Depression; HDS-R revised Hasegawa’s Dementia Scale; HKLLT Honk Kong List Learning Test; HOA healthy old adults; HYA healthy young adults; J-ZBI Japanese version of the Zarit Burden Interview; MMSE Mini-Mental State Examination; MoCA Montreal cognitive assessment; NPI Neuropsychiatric Inventory; PFC prefrontal cortex; RCFT Rey-Osterrieth Complex Figure Test; RWT Regensburg Word fluency Test; STT Shape Trail Test; VLMT Verbal Learning and Memory Test; VSRAD voxel-based specific regional analysis system for Alzheimer’s disease; TAP Test of Attentional Performance; TCD transcranial doppler; WMS-R revised Wechsler Memory Scale; WMT working memory task; (*) control group is not age-matched with patient group; (**) cross-sectional study with multiple fNIRS acquisitions.

**Table 3 diagnostics-10-00581-t003:** Summary of peer-reviewed articles of fNIRS applications in Multiple Sclerosis (MS) patients found trough literature search of NCBI PubMed and Web of Science databases.

	Multiple Sclerosis (MS)							
	fNIRS Device	Patient Info	Study Type	Integrated Clinical Outcomes	Investigated Cortical Areas	Performed Task	Ch.	Multimodal Integration
**Chaparro et al., 2017**	fNIR Imager 1000 (fNIR Devices LLC, Protomac, MD, USA)	10 MS, 12 HC	cross-sectional	RBANS, SPPB	PFC	walking while talking with/without partial body weight support (WT, DWT)	16	instrumented treadmill
**Saleh et al., 2018**	NIRSport (NIRx Medizintechnik GmbH, Germany)	14 RRMS, 14 HC	cross-sectional	BVMT-R, CVLT-II, SDMT, T25FW	bilateral premotor and supplementary motor areas	serial 7′s subtraction cognitive task, walking and DWT	12	no
**Hernandez et al., 2016**	fNIR Imager 1000 (fNIR Devices LLC, Protomac, MD, USA)	8 MS, 8 HC	cross-sectional	EDSS	PFC	walking without/while talking (WT, DWT)	16	instrumented walkway
**Hernandez et al., 2019**	fNIR Imager 1000 (fNIR Devices LLC, Protomac, MD, USA)	10 MS, 12 HC (*)	cross-sectional	n.d.	PFC	reciting alternate letters, virtual beam walking without/while talking	16	instrumented treadmill
**Borragán et al., 2018**	BrainSight NIRS V2.3b16 (Rogue Research Inc., Canada)	10 RRMS, 11 HC (*)	cross-sectional	VASf	dorsolateral and ventrolateral PFC, inferior parietal cortex	TloadDback (dual working memory task)	24	polysomnography (including EEG, EOG, EMG, ECG, abdominal and thoracic belts, oronasal airflow and thermocouple, finger pulse oximeter)
**Jimenez et al., 2014**	CW5 (TechEn Inc., Milford, MA, USA)	4 RRMS, 2 SPMS, 2 MS, 1 CIS, 1 MS (unspecified), 8 HC	cross-sectional	n.d.	motor cortex	unilateral finger tapping, resting-state	n.d.	no
**Wolff et al., 2019**	NIRSport (NIRx Medical Technologies LLC, NY, USA)	26 RRMS, 18 SPMS, 6 PPMS	cross-sectional	BDI, CR10 scale, FSMC, SCS-K-D	PFC (BA10)	isometric contraction task	22	Hand dynamometer
**Stojanovic-Radic et al., 2015**	DYNOT Imaging System, Model 264 (NIRx Medical Technologies LLC, Glen Head, NY, USA)	13 MS, 12 HC (*)	cross-sectional	n.d.	bilateral superior frontal and middle frontal gyri (BA10)	n-back task (working memory task)	900	no
**Molinari et al., 2014**	NIRO-200 (Hamamatsu Photonics K.K., Japan)	22 RRMS, 10 other neurological disease, 22 HC (*)	cross-sectional	n.d.	FC	monitoring during ozone autohemotherapy	2	transcranial doppler, ozone autohemotherapy
**Molinari et al., 2017**	NIRO-200 (Hamamatsu Photonics K.K., Japan)	10 RRMS, 10 HC (*)	cross-sectional	n.d.	bilateral FC	monitoring during ozone autohemotherapy	2	ozone autohemotherapy
**Yang and Dunn, 2015**	ISS OxiplexTS (ISS Inc., Champaign, IL, USA)	51 RRMS, 16 SPMS, 15 PPMS, 3 CIS, 19 HC	cross-sectional	EDSS, SDMT, T25FT, 9 hole peg test	bilateral FC	tissue oxygenation monitor	4	concurrent NIRS-MRI

BDI Beck’s Depression Inventory; BVMT-R Brief Visuospatial Memory Test-Revised; CIS clinically isolated syndrome; CR10 category ratio 10; CVLT-II second edition of California Verbal Learning Test; DWT dual walking task; EDSS Expanded Disability Status Scale; FC frontal cortex; FSMC Fatigue Scale for Motor and Cognitive Functions; PFC prefrontal cortex; PPMS primary progressive multiple sclerosis; RBANS Repeated Battery for the Assessment of Neuropsychological Status; RRMS relapse remitting multiple sclerosis; SCS-K-D German adaptation and short form of Self-Control Scale; SDMT Symbol Digit Modalities Test; SPMS secondary progressive multiple sclerosis; SPPB Short Physical Performance Battery; T25FW timed 25-foot walk test; VASf Visual Analog Scale for fatigue; WT walking task; (*) control group is not age-matched with patient group.

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
