# Peer review of "A Systematic Review of Cerebral Functional Near-Infrared Spectroscopy in Chronic Neurological Diseases—Actual Applications and Future Perspectives"

_diagnostics, 2020, doi:10.3390/diagnostics10080581_

Round 1

Reviewer 1 Report

In this review article, the authors cover the application of functional near infrared spectroscopy (fNIRS) in chronic neurological diseases. I think this is a very comprehensive and thorough paper, so I congratulate the authors for their work. I have the following suggestions that I think would improve the manuscript.

fNIRS is becoming mature and is definitely a valuable tool in neuroscience. Recently, optoacoustic (photoacoustic) methods synergistically combining optical contrast and ultrasound resolution have been shown to overcome the resolution limitations of optical methods. Optoacoustics has successfully been used in neuroimaging applications involving small animals (see e.g. Gottschalk et al. Neurophotonics, 4(1), 011007 (2016), Cao et al. Neuroimage, 150, 77-87 (2017), Mc Larney et al. Frontiers in Neuroscience, 14, 536 (2020), Ovsepian et al. Photoacoustics, 17, 100153 (2020)). Potentially, optoacoustics can also be applied in the clinics to assess neurological disorders. Although this has not been achieved, I think it is important to mention it in the discussion. Additionally, optoacoutics has been shown to be sensitive to calcium and voltage indicators as well as hemodynamic changes, which can potentially serve to elucidate the mechanisms of neurovascular coupling (see e.g. Rao et al. Scientific reports, 7(1), 1-10 (2017), Gottschalk et al Nature biomedical engineering, 3(5), 392-401 (2019), Degtyaruk et al. Photonics (Vol. 6, No. 2, p. 67) (2019)).

Related to the previous comment, optoacoustics and fNIRS can potentially be combined to overcome the limitations of each other, where optoacoustics can serve to enhance the limitations of fNIRS and fNIRS can be used to provide quantitative information to be used in optoacoustics. This could also be mentioned.

The authors mention that continuous wave fNIRS is the main solution in actual instrumentations, while frequency domain and time domain fNIRS are only so widely used. Actually, I think frequency domain and time domain fNIRS are currently used more frequency than continuous wave fNIRS. I would suggest to include references to substantiate this.

Check for typos, “World Health Organization” instead of “Worldwide Health Organization”,… I would also avoid using informal expression such as “anyhow”.

Author Response

Dear Reviewer,

Thank you for your interesting comments.

We have improved the ‘Discussion’ following your suggestion about the investigation of neurovascular coupling mechanisms underlying the potential benefits of multimodal integration also with respect of the actual advantages of photoacoustic related to the drawbacks of traditional fNIRS.

Concerning the main solution in actual instrumentations, the respective phrase in the ‘Introduction’ was extended. CW-fNIRS still occupies a larger portion of actual applications due to the presence of a higher number of commercial systems, as reported both by Torricelli et al. [1] and more recently by Fantini and Sassaroli [2]. For this reason, FD- and TD-fNIRS are more related to the development of new optical brain imaging methods rather than actual applications, as illustrated also in the actual PD, MS and AD/MCI sections of this paper. However, FD- and TD-fNIRS represent promising technology that will compensate the drawbacks of CW-fNIRS and possibly substitute them in future applications.

[1] A. Torricelli et al., “Time domain functional NIRS imaging for human brain mapping,” Neuroimage, vol. 85, pp. 28–50, Jan. 2014.

[2] M. Ferrari and V. Quaresima, “A brief review on the history of human functional near-infrared spectroscopy (fNIRS) development and fields of application,” NeuroImage. 2012.

Reviewer 2 Report

The manuscript is a comprehensive, well written and well discussed systematic review on the use  of cerebral functional Near- Infrared Spectroscopy in chronic neurological diseases.

The applicability of the technique in these pathologies make the manuscript of  potential interest for a wide audience of readers.

Comment

The inclusion in the tables of   brand /manufacturer of the device used in each study may represent an useful completion of the overview.

Author Response

Dear Reviewer,

Thank you for your comments.

Following your suggestions, we have included in all the tables an additional column named “Device” which indicates the model and manufacturer of the employed fNIRS system of each reviewed study.